# Utilising Hyperspectral Autofluorescence Imaging in the Objective Assessment of Disease State and Pain in Patients with Rheumatoid Arthritis

**DOI:** 10.3390/ijms252211996

**Published:** 2024-11-08

**Authors:** Florence Lees, Saabah B. Mahbub, Martin E. Gosnell, Jared M. Campbell, Helen Weedon, Abbas Habibalahi, Ewa M. Goldys, Mihir D. Wechalekar, Mark R. Hutchinson, Tania N. Crotti

**Affiliations:** 1School of Biomedicine, University of Adelaide, Adelaide, SA 5005, Australiatania.crotti@adelaide.edu.au (T.N.C.); 2ARC Centre for Excellence for Nanoscale Biophotonics, University of Adelaide, Adelaide, SA 5005, Australia; 3Graduate School of Biomedical Engineering, University of New South Wales (UNSW), Sydney, NSW 2052, Australia; saabah.mahbub@gmail.com (S.B.M.); j.campbell@unsw.edu.au (J.M.C.); a.habibalahi@gmail.com (A.H.); 4ARC Centre of Excellence for Nanoscale Biophotonics, University of New South Wales (UNSW), Sydney, NSW 2052, Australia; 5Quantitative Pty Ltd., 118 Great Western Highway, Mount Victoria, NSW 2786, Australia; martingosnell@gmail.com; 6Department of Rheumatology, Flinders Medical Centre and Flinders University, Bedford Park, SA 5042, Australia; weedonh@gmail.com (H.W.);

**Keywords:** arthritis, hyperspectral, rheumatoid, inflammation

## Abstract

Rheumatoid Arthritis (RA) is a chronic inflammatory disease resulting in joint swelling and pain. Treatment options can be reliant on disease activity scores (DAS) incorporating patient global assessments, which are quantified via visual analogue scales (VAS). VAS can be subjective and not necessarily align with clinical symptoms, such as inflammation, resulting in a disconnect between the patient’s and practitioners’ experience. The development of more objective assessments of pain would enable a more targeted and personalised management of pain within individuals with RA and have the potential to improve the reliability of assessments in research. Using emerging light-based hyperspectral autofluorescence imaging (HAI) technology, we aimed to objectively differentiate disease and pain states based on the analysis of synovial tissue (ST) samples from RA patients. In total, 22 individuals with RA were dichotomised using the DAS in 28-joint counts (DAS-28) into an inactive (IA) or active disease (active-RA) group and then three sub-levels of pain (low, mid, high) based on VAS. HAI was performed on ST sections to identify and quantify the most prominent fluorophores. HAI fluorophore analysis revealed a distinct separation between the IA-RA and active-RA mid-VAS cohort, successfully determining disease state. Additionally, the separation between active-RA Mid-VAS and active RA High-VAS cohort suggests that HAI could be used to objectively separate individuals based on pain severity.

## 1. Introduction

Rheumatoid Arthritis (RA) is a chronic inflammatory autoimmune disease resulting in joint pain, swelling, and stiffness [1,2]. Pain, once thought to be driven via inflammation, has now been theorised to also be associated with non-inflammatory causes [3]. A disconnect between the presence of inflammation and pain experienced by the patient can lead to difficulties for the clinician when making crucial decisions relating to the best treatment path to take. Therefore, further research into why this disconnect occurs between pain state and RA disease activity level is required. The ineffectiveness of treatments to curb heightened pain levels has resulted in dissatisfaction [4] and the inability to continue full-time work in patients [5]. Therefore, RA presents not only a health but also an economic burden.

Disease activity in RA is tracked using a set of composite measures alongside imaging such as the disease activity score utilising a 28-joint count (DAS-28) in which the clinician assesses the number of swollen and tender joints along with inflammatory markers; for example, erythrocyte sedimentation rate (ESR) or the C-reactive protein (CRP) and patient global assessment (PGA) of disease activity [6]. PGAs are subjective patient-self-reported assessments of disease activity that are heavily influenced by pain and can misalign with clinical symptoms, such as inflammation [4,7]. The visual analogue scale (VAS) is an example of a self-reported assessment of pain that has reported limitations [8,9]. Thus, there is a need to introduce more objective measures for RA disease activity and/or pain and to differentiate pain that is caused by inflammatory versus non-inflammatory mechanisms. Such differentiation is critical for therapeutic decision-making with regard to the potential escalation of immunosuppressive disease-modifying treatment and/or the addition of treatments directed to pain rather than immunosuppression-based approaches. Additionally, the improved reliability of objective assessments could help to reduce data heterogeneity, accelerating research into clinical interventions as well as attempts to understand the disease course of RA.

Hyperspectral autofluorescence imaging (HAI) involves the use of a spectrum of light (colours) to create images capturing spectroscopic as well as morphology information, which are highly informative (Figure 1). HAI utilises the endogenous molecular fluorophores of individual cells and tissue to fingerprint the sample. Endogenous fluorescent molecules such as nicotinamide adenine dinucleotide phosphate (NADPH), flavin adenine dinucleotide (FAD), flavin mononucleotide, and collagen levels in cells and tissues are modified during physiological processes [10]. The assessment of cell and tissue autofluorescent spectra can enable a broad range of morphology and structural features to be identified [11,12]. For example, differences in metabolic signatures were observed via HAI within the lumbar spinal cord of rats with chronic constriction injuries [13]. This suggests that complex mechanisms, such as changes in pain states, elicit spectral changes in tissue that can be measured using HAI.

Biophotonic technology has been used in human and mouse models of disease. Optical coherence tomography angiography (OCTA), which, like HAI, relies on light waves rather than dyes, has been used to monitor retinal microvascular density in hydroxychloroquine-related retinal toxicity in RA patients [14], while in a collagen antibody-induced (CAI) mouse model, photoacoustic (PA) imaging was able to distinguish early changes that were not detected by clinical observation [15]. Dual modality multiparametric PA and ultrasound imaging allow a single platform integration of both photoacoustic and ultrasound imaging and have been used to assess distal interphalangeal joints in a rat adjuvant arthritis model and healthy patients [16]. In this work, Guo et al. were able to demonstrate the effectiveness of RA treatment via quantitative hemodynamic and morphological parameters. Dual PA/ultrasound imaging allows a single platform integration of both photoacoustic and ultrasound imaging, and in a CAI model [17], this method was able to identify subtle changes, which may be missed within the early stages of RA disease. This highlights the potential of biophotonic technology in medical research and clinical practice.

**Figure 1 ijms-25-11996-f001:**
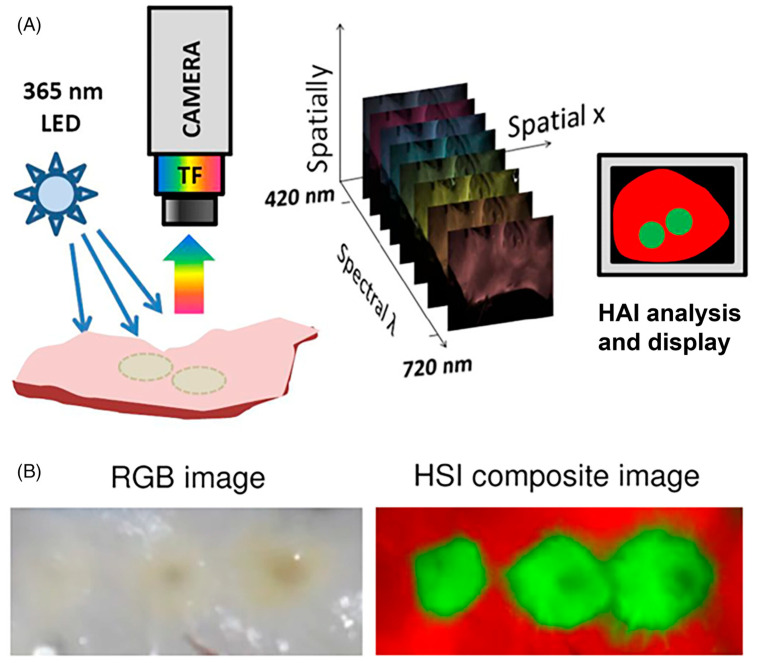
(**A**), Steps of hyperspectral image acquisition and processing. A sample is illuminated by a given wavelength and emitted light resulting from the excitation is collected through filters coupled to a camera. A three-dimensional data stack (*x*,*y*,*λ*) comprises *x*-*y* images at different wavelengths (*λ*), which can then be interpreted for display. (**B**) Side-by-side comparison of an endocardial left atrial surface with three radiofrequency ablation under room light and a composite hyperspectral image of the same tissue, which clearly shows the borders of the lesion. *Source*: Adapted from Muselimyan et al. [18] under a cc attribution license.

In our study, synovial tissue (ST) samples were isolated from patient cohorts consisting of inactive RA patients with low- and mid-levels of pain, and active RA patients with mid and high levels of pain. Here, we hypothesise that there are molecular signatures in the target tissue of RA, the ST, which are sufficient to differentiate disease state and pain severity. We aimed to (a) objectively determine the disease state (active versus inactive), regardless of pain severity, as well as for the same pain severity; (b) differentiate the degree of pain (as measured by VAS), separately in active and inactive RA; (c) examine if these signatures can be detected with HAI of ST; and (d) identify whether quantitative analysis of the HAI images can identify regions of the issue where these key signatures originate.

## 2. Results

### 2.1. Patient Demographic

The study included ST samples from 22 participants with RA (13 female, 9 male). In the inactive group, the mean DAS28 was 1.8, while for the active group, it was 6.3. The mean VAS pain score was 9.0 and 65.5 for patients with inactive and active RA, respectively, while AS fatigue was 14.7 and 59.9. The average age of the cohort was 55.4 ± 9.4 years. In the inactive group, 4 out of the 9 participants were female, while in the active group, it was 9 out of 13. Participants fulfilled the 2010 American College of Rheumatology (ACR)/European Alliance of Associations for Rheumatology (EULAR) criteria [19].

### 2.2. Histological Assessment of Inflammation Within the ST

Representative images of CD68 staining within the ST sublining of an inactive RA and active RA patient’s sample are presented in Figure 2A,B. Samples used in the representative images were categorised into active (Figure 2A) versus inactive (Figure 2B) RA based on the DAS-28 criteria based on their DAS-28 score (Figure 2C). Samples were then categorised based on their CRP scores as a marker of inflammation (Figure 2D). There was no statistically significant difference between the CRP scores for the inactive RA patients when compared to the active RA cohort (Figure 2C,D).

### 2.3. Differentiation of RA Disease States and Pain States Within Each from Synovial Tissue Cells Only

Autofluorescence signatures obtained from the segmented cells in inactive and active-RA patients were analysed using the abundances of the unmixed fluorophore and the optical redox ratio. Unmixing detected three main fluorophore groups, (1) NAD(P)H comprising NADH pure and protein-bound and NADPH pure and protein-bound; (2) flavins, pure and bound; and (3) collagens. The results show that an increased level of NAD(P)H was observed within the active disease groups compared to inactive groups (Figure 3A). The NAD(P)H in the RA-Mid group was significantly increased compared to all other groups (*p* < 0.01). A decreased level of flavins (Figure 3B) was found in the RA-High group compared to the inactive patient. The inactive RA-Mid group has shown a significant increase in flavins’ abundance compared to all other RA groups (*p* < 0.001). Collagen abundance was significantly higher in the RA-High group compared to all other groups (Figure 3C) (*p* < 0.001). A significant change in the optical redox ratio between inactive and active RA patients (Figure 3D) was observed; the optical redox ratio was significantly higher in the active RA groups compared to all other inactive RA groups (*p* < 0.001)

Fluorophore abundances and their redox ratio can be treated as quantitative patient features. Such feature analysis in SC was also able to separate patients with inactive and active RA (Figure 4A). These four biologically interpretable features yielded a clear separation of these RA disease states (Appendix A). Pain-related groups in each RA disease state were also distinguished by using similar features. This was observed separately in inactive RA (Figure 4B; Appendix A) and active RA (Figure 4C; Appendix A). The separation between inactive-Mid and RA-Mid patients was also observed (Figure 4D; Appendix A). The resulting ROC curves with AUC calculations are included in the Appendix A.

### 2.4. Differentiation of RA Disease States and Pain States Within Each from Synovial Tissue Fibres Only

A similar analysis for the cells (fluorophore unmixing and feature analysis) was conducted for ST fibres. The unmixing of fluorophores detected an increased level of NAD(P)H within the active-RA disease state compared to inactive-RA (Figure 5A). A decreased level of flavins (Figure 5B) was observed in the RA-High group compared to RA-Mid; however, NAD(P)H in the RA-Mid and RA-High groups were higher than in the inactive group (IA-Low and IA-Mid). The IA-Mid group had a significantly greater abundance of NAD(P)H compared to all other groups (*p* < 0.001). Collagen abundance was (Figure 5C) significantly higher in the RA-High group compared to the IA-Low and RA-Mid groups (*p* < 0.001). A significant increase in the optical redox ratio was measured in the RA-High group compared to all other groups (*p* < 0.05 for IA-Mid and RA-Mid groups, *p* < 0.001 for IA-Low) (Figure 5D).

The results show that the HAI patterns of fibres could differentiate the RA disease state (Figure 6A). To differentiate between inactive and active RA, four of our four previously described biologically interpretable features were sufficient (Appendix A). The differentiation of pain-related groups separately in each RA disease state, for both inactive RA (Figure 6B) and active RA (Figure 6C), was also observed. In this analysis, three or four features (Appendix A) were utilised. Similar analysis (Figure 6D; Appendix A) shows a clear separation (AUC = 0.947) between inactive-Mid and active RA-Mid patients. The respective ROC curves with AUC are included in the Appendix A.

### 2.5. Deep Learning Encoder Algorithm for the Identification of Inflammatory Signatures

The imaging of inactive and active-RA signatures has been carried out by compressing complex multidimensional HAI data into 4 colour (CMYK) images (Figure 7A–D). This was carried out in a way that optimally reveals the differences between patients for both inactive and active disease groups (Appendix A). Colour differences were observed, for example, IA-low (blue) and RA-high (purple). Slight differences were also observed between groups with different pain scores in each RA disease state.

### 2.6. Collagen II Staining of the ST to Compare Relative Abundance Levels of Collagen from HAI

The active-RA-high cohort appeared to have the highest average integrated overall density levels within ST; however, this was not a significant difference compared to all other groups (Figure 7E).

## 3. Discussion

This study investigated the use of HAI to differentiate between disease and the pain state, utilising a human cohort consisting of patients with both active and inactive RA, as well as a wide spectrum of VAS pain scores from 1 to 100. The histopathological features of inflammation of the ST were assessed via IHC staining for CD68, a marker of macrophages [20]. ST inflammation was scored using an established semi-quantitative method [21]. Patients were initially categorised based on the DAS-28 score, resulting in a higher CD68 score for inactive patients compared to the CD68 score in active RA. This was not consistent with previous findings where RA patients treated with prednisolone had concomitant reductions in CD68 and DAS-28 [22]. The observation probably reflects the inexact correlation between the DAS28, a clinical–epidemiological construct that is heavily influenced by patient perception, vs. local synovial pathology, including intrinsic variability between areas of inflammatory infiltrate, which may influence CD68 scores. The incorporation of a PGA-VAS (significantly influenced by the patient’s pain perception [8,9]) within the DAS-28 score possibly caused a disconnect between what was observed in the tissue and what was calculated via the DAS-28 formula and demonstrated the inherent limitations of the DAS-28 tool. Taking this possibility into account, patients were then separated based on measurements of CRP, an objective inflammatory marker [23]. Although not statistically significant at *p* < 0.05, an increase in the CD68 staining score within an ST of patients with active-RA compared to inactive patients was observed when patients were separated based on CRP categorisation. This supports CRP as a more accurate measure of inflammation without being confounded by the pain component.

The ability of HAI to differentiate RA status (grouped using DAS28) and pain (grouped by VAS) was investigated in both the SC and FT of the synovium. Autofluorophore features in FT had better separability characteristics compared to cellular analysis. Within FT, there was a clear separation between autofluorophore features quantified in the RA-Mid and RA-High groups as well as between IA-Mid and RA-Mid. However, there was no differentiation between IA low and mid-pain levels. This might suggest that a lower level of disease activity observed in an inactive case might reduce our ability to differentiate pain states.

Autofluorescent feature-based separation being more prominent in FT rather than SC is a novel finding as the clinical standard has been to investigate fibroblast-like synoviocytes (FLS) only [24]. FLS has been shown to be a key component involved in the pathogenesis of RA [24]. Our findings suggest that FT may also play a key role or reflect the molecular signature that is associated with the perception of pain in patients with a mid-level of pain.

The relative abundance of NAD(P)H and flavins was previously used to quantify tissue differences [12]. In this work, following unmixing, auto-fluorophore spectral signals of NAD(P)H, flavins, and collagen were measured in both the cellular and fibrous data sets. NAD(P)H relative amounts were found to be highest in patients with active RA for both FT and SC compared to inactive RA. This is consistent with an investigation on NADPH oxidases in FLS from patients, which showed an increase in reactive oxygen species in inflammatory arthritis RA compared with non-inflammatory forms, such as osteoarthritis [25]. These findings suggest that there are metabolic changes occurring specific to the pathogenesis observed in RA compared to other forms of arthritis. These results align with the literature, which reports that those metabolic changes are observed in OA [26] and RA [27].

Within the cellular and FT data set, a decrease in flavin within patients with active RA was measured. A depletion of riboflavins has been previously reported to occur more within active-RA patients than inactive patients compared to healthy controls and with patients measuring increased pain levels [28]. The current study found similar results in that the lowest flavin hyperspectral quantification was associated with the active-RA high-VAS cohort. We found that within FT, the highest amount of flavins was measured within the active-RA mid-VAS cohort. Interestingly, within complete Freunds adjuvant (CFA) mice, flavins were linked to a reduction in the production of TNF-alpha, a pro-inflammatory cytokine [29]. This might suggest that the patients with mid-level VAS pain had higher levels of flavins contributing to their slightly lower pain state.

The abundance ratio of NAD(P)H to flavins was measured to produce the (optical) redox ratio, established to be a measure of metabolic and mitochondrial activity [30]. The highest redox ratios were found here in patients who had active RA in both FT and SC data sets. Our findings are similar to a study where FLS obtained from patients with RA had increased metabolic changes in sugar metabolism, lipolysis, and amino acid metabolism compared to samples obtained from patients with osteoarthritis (OA) [31]. Another study reported that the metabolic profile of FLS from healthy control and early RA were distinct enough to identify groupings [32]. The increase in pro-inflammatory cytokines in RA may alter cell metabolism in inflamed joints (reviewed by [33]). The metabolic changes in glycolytic and mitochondrial metabolic pathways discussed within this review are associated with synovial hyperplasia and inflammation and are targets of clinical trials.

Within the body, there are several types of collagens, most notably, types 1, 2, and 3. Type 1 is most abundant in bones, type 2 within articular cartilages, and type 3 within reticular fibres, which are involved in supporting soft tissue. Each type is found in differing abundance depending on body location. In the current study, the relative abundance of collagen was assessed via HAI; however, the separation of specific collagen types was not possible due to spectral overlapping. Differentiating between the three types would allow a more specific analysis of the differences between the differing disease and pain groups. Both data sets produced similar results with IA-Low having the lowest level of collagen and RA-High having the highest level of collagen. Simultaneously, ST was also stained fluorescently for anti-collagen II antibodies. Using a known standard stain ensured that we could compare data obtained from fluorescent imaging and HAI to assess how closely they aligned. A close alignment was observed between the fluorescent anti-collagen II antibody stain compared to relatively abundant levels measured from HAI unmixing, which supports our collagen identification. 

As RA disease states could be separated using HAI, the location in the tissue of the key spectral characteristics that were generating the critical “disease state signatures” was of interest. To establish this, machine learning was applied to BF images obtained from the hyperspectral microscope, following the approach in [13]. From this, the autofluorescent signatures highlighting the disease state were captured, allowing the heterogeneous composition of the tissue to be visualised. Future studies should explore the use of this type of hypothesis-generating technology to explore previously unrecognised contributors to disease states.

Limitations specific to the use of unsupervised machine learning analysis include having a small heterogeneous sample set. Hence, this should be considered as a training data set that requires further testing to examine how the model will generalise to additional data sets. A related limitation is that we were unable to divide activity statuses with greater granularity, which would enable increased precision of findings, nor were sub-group analyses (e.g., females in comparison to males) possible. A longitudinal clinical cohort study with a larger sample size is required to further validate the use of HAI in RA management.

Within the FT, it was possible to differentiate both disease and pain states via feature analysis. HAI may help further objectify pain assessment in patients. This would be highly significant in the clinical trial sphere to ensure that treatments are assessed effectively. Traditional studies have used subjective measures, such as PGA VAS, to assess whether a treatment has worked. However, this has the disadvantage that patients may clinically respond (e.g., with a reduction in their swollen joint counts and inflammatory markers) but still not detect a difference (because of ongoing pain impacting their perception of disease activity and the visual analogue score for the patient global assessment), therefore lowering the perceived effectiveness of the treatment on pain. There is also the potential for confounding from co-morbidities such as fibromyalgia. Our findings provide promising support for the feasibility of HAI technology for management and pain assessment, although conventional examination through validated methodologies remains vital. Future research could further examine if the signal differences detected relate to pain via examination of synovial biology. This points to a future of precision medicine in RA. Furthermore, the extension of this work to animal models could enable objective assessment of pain in a context where even subjective quantification can be a challenge.

## 4. Materials and Methods

### 4.1. Patients

The ST samples from 22 (13 female, 9 male) participants with RA (per the 2010 American College of Rheumatology (ACR)/European Alliance of Associations for Rheumatology (EULAR) criteria) [19] from the Flinders Medical Centre were utilised for this study. All ST samples were obtained by a mini arthroscopy [34]. The data obtained from patients included demographics, physician and patient global disease assessments, and pain and fatigue VAS levels (on a scale of 0–100), health assessment questionnaire (HAQ), joint counts, and inflammatory markers (C-reactive protein and erythrocyte sedimentation rate (ESR)). Relevant metrics (joint counts, ESR, and global assessments) were utilised to generate the DAS28 (the DAS28 using the ESR was used for this study). All methods were carried out in accordance with relevant guidelines and regulations [35]. All experimental protocols were approved by the Southern Adelaide Local Health Network Human Research Ethics (HREC; 199.10, 396.10) and The University of Adelaide’s Human Research Ethics (H-35.2001) committees. Written informed consent was obtained from participants before sample collection.

### 4.2. Study Design

The ST samples obtained from each of the 22 participants were initially split into an inactive RA group (*n* = 9) or active RA group (*n* = 13) and then further divided according to their self-reported pain VAS: low (1–10) mid (11–59) and high (60–100). The clinical characteristics of the patient cohorts and their disease can be found in the (Appendix A). The groups in this study are referred to herein as inactive-low (IA-Low, *n* = 8), inactive-mid (IA-Mid, *n* = 1), active-mid (RA-Mid, *n* = 4), and active-high (RA-High, *n* = 9). There were no individuals within the inactive-high or active-low group, so only the 4 above groups were investigated. Inactive individuals had a DAS28 ≤ 2.6 while active individuals had a DAS28 > 2.6 [35]; in addition, samples were classified according to the CRP: CRP-inactive individuals had a level < 8 mg/L while active had a level > 8 mg/L.

### 4.3. Histological Assessment of Synovial Tissue (ST)

Immunohistochemical detection of CD68 and type II collagen was conducted on acetone-fixed fresh frozen 5 µm-sectioned ST sections.

### 4.4. CD68 Staining

ST sublining CD68 expression has been identified as a benchmark when examining tissue for clinical treatment as it is an observed biomarker of response to treatment [20]. Sections were washed in 1× PBS (pH 7.4, 3 × 5 min) and then incubated with blocking buffer (0.1% NaN_3_/1% H_2_O_2_/TPBS) for 20 min at room temperature. Following a 1 × PBS wash, sections were blocked with 20% normal donkey serum for 30 min at room temperature. Sections were incubated with primary antibody (EBM11 1:400 Catalogue #M0718, Dako Australia), diluted in 1 × PBS/1% BSA overnight at 4 °C, and washed in 1 × PBS. Secondary goat anti-mouse antibodies (HRP 1:100 Catalogue #P0447; Dako, Denmark, SC, USA) diluted in 1 × PBS 1% BSA/10% NHuS were incubated for 30 min at room temperature, followed by 1 × PBS wash. Sections were then incubated with tertiary rabbit anti-goat antibody (HRP 1:100 catalog #P0449; Dako, Adelaide, Australia) for 30 min at room temperature, followed by 1 × PBS wash. Sections were incubated with substrate chromogen (AEC ready-to-use #K3469; Dako Australia), counterstained with haematoxylin, and then mounted with Aquatek. CD68 positive cells were assessed within a 0.68 mm region of interest (ROI) within the sub-lining by two blinded observers as per a previously published semi-quantitative scoring method [21].

### 4.5. Type II Collagen Staining

Sections were washed and then incubated with a blocking buffer for 60 min at room temperature. Primary antibody diluted in blocking buffer was added and incubated overnight at 4 °C and washed. Donkey anti-rabbit secondary antibody was diluted in 1 × PBS and added to sections before incubation for 60 min at room temperature and washed. The intensity of staining was measured using the software Fiji (Image J version 2.0).

### 4.6. Hyperspectral Autofluorescence Imaging and Bright-Field Imaging (HAI)

ST was freshly frozen in optimal cutting temperature compound and stored in a −80 °C freezer. Then, 5 µm thick sections were cut using a cryostat and placed on a coverslip. Tissue sections were warmed at 37 °C in Hank’s Balanced Salt Solution (HBSS) to prevent excess drying. An adapted standard 40× silicone U12TM objective fluorescence microscope (Olympus iX83™ (Olympus, Tokyo, Japan)) with 69 fluorescent channels (Appendix A) was used in HAI to image sample autofluorescence [36,37,38]. Each HAI data set had an associated bright field image. 

### 4.7. HAI Data Analysis

Background fluorescence, dead or saturated camera pixels, Poisson’s noise, and flattening of uneven illumination to minimise image artefacts [11,12,38] were performed on HAI data during pre-processing. Segmentation was performed on the ST bright field image to individually isolate the regions of each synoviocyte cell (SC) and collagenous fibre tissue (FT) structure. These regions of interest (ROIs) were then used for further analysis. See Appendix A for further details.

### 4.8. Unmixing of Endogenous Fluorophores

Unmixing is the process of extracting spectral characteristics of dominant endogenous fluorophores from the total autofluorescence signal for each sample on a pixel-by-pixel basis. Linear mixing model-based unmixing was used to calculate the relative abundance (normalised cellular concentrations) for each extracted fluorophore [11,39]. The unsupervised unmixing method Robust Dependent Component Analysis (RoDECA) [38] was employed to extract the native fluorophores’ spectra and calculate their relative abundance from the pre-processed data.

### 4.9. Analysis of Cellular and Fibre Features and Classification

The autofluorescence features from the segmented cells and fibres were calculated to evaluate how well they differentiate the patient groups. These features included the mean abundance of unmixed fluorophores [11,39,40,41] (Appendix A). Of note is the optical redox ratio calculated here as the abundance ratio of NAD(P)H to flavins [42,43]. Four autofluorescent features for each segmented cell/fibre were calculated. Canonical discriminative analysis was then used to project the feature data from the pairwise study groups onto an optimal two-dimensional (2-D) space, which minimized within-group variance while maximizing the between-group distance [44]. This type of analysis produces canonical variables given by selected linear combinations of cellular features (see axes in Figure 4 and Figure 6). Furthermore, a simple linear classifier [45] was used to predict the pre-determined patient groups from the HAI alone, without clinical knowledge. A receiver operating characteristic curve (ROC) for this classifier was then generated to quantify how well the HAI data can render our study groups. The area under the curve (AUC) was calculated to characterise the classifier performance.

### 4.10. Deep Learning Autoencoder Algorithm to Facilitate Imaging of Group Signatures

A deep learning algorithm based on an autoencoder was developed to create an expertly interpreted version of HAI images that highlight the patient group signatures obtained from ST [13]. This algorithm compresses the highly dimensional HAI images into exactly four colour (CMYK) variables, enabling the clear visualisation of regions where sample autofluorescence was modified by molecular changes within the considered patient groups. See Appendix A for more details.

### 4.11. Statistical Analysis

Statistical analysis utilised GraphPad Prism^®^ software (V7.03; GraphPad Software, La Joella, CA, USA) for immunohistochemistry and immunofluorescence data sets or MATLAB 2019b for extracted fluorophores within cells and tissue fibres. The non-parametric Mann–Whitney U test was used to assess for significance of the observed differences between groups. Differences were determined to be significant if *p* < 0.05.

## 5. Conclusions

For the first time in human ST, this study demonstrates that (a) HAI can be utilised to discern between tissues from a patient with RA in an active versus inactive disease state and that (b) HAI can be used to detect a significant separation between VAS pain groupings. Our results provide the first evidence of the potential utility of this novel technique to stratify RA disease state and VAS pain groupings, potentially fulfilling the unmet need for an objective measure of pain to guide clinical decision-making in those patients with RA who appear to be clinically quiescent but suffer from ongoing pain.

## Figures and Tables

**Figure 2 ijms-25-11996-f002:**
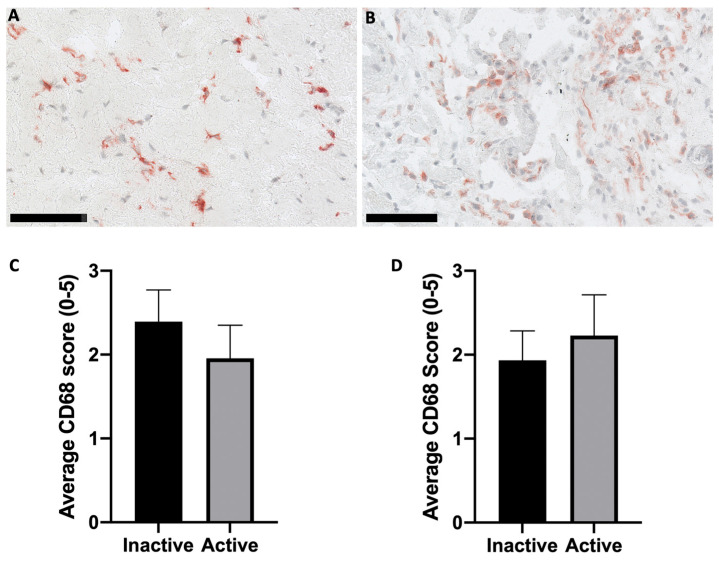
Histological assessment of CD68 within ST. (**A**) Inactive RA disease state. (**B**) Active RA disease state. Representative samples used for images (**A**,**B**) were categorised based on the DAS-28 score. Representative images were obtained at 20× magnification. Semi-quantitative analysis of CD68 positive cells was performed and then categorised based on the (**C**) DAS-28 and (**D**) CRP values. Scale bars are representative of 100 µm. No statistically significant difference was observed in either grouping method. Error bars are SEM.

**Figure 3 ijms-25-11996-f003:**
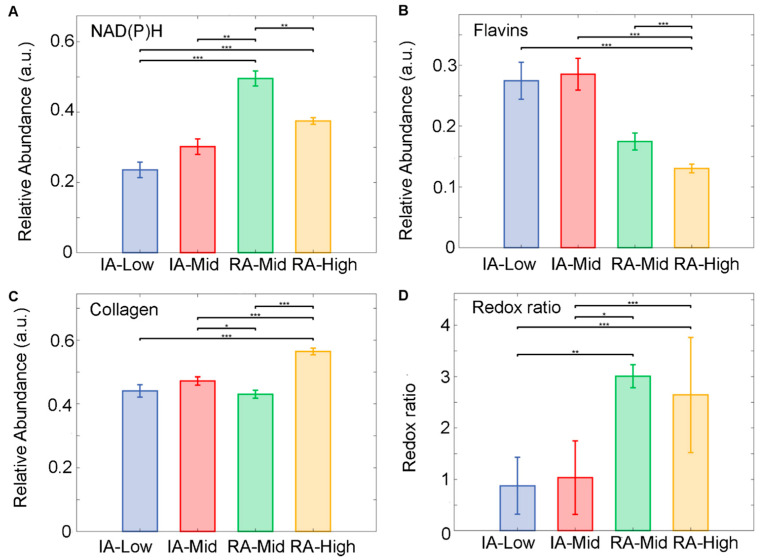
Unmixed auto fluorophore spectral signals from SC within ST samples obtained from inactive (IA) and active-RA patients. (**A**) Relative abundance of NAD(P)H, (**B**) relative abundance of Flavins, (**C**) relative abundance of collagen, and (**D**) the ratio of relative abundances of NAD(P)H to Flavin’s. Median of the relative abundances displayed as SEM. * *p* < 0.05, ** *p* < 0.01 and *** *p* < 0.001.

**Figure 4 ijms-25-11996-f004:**
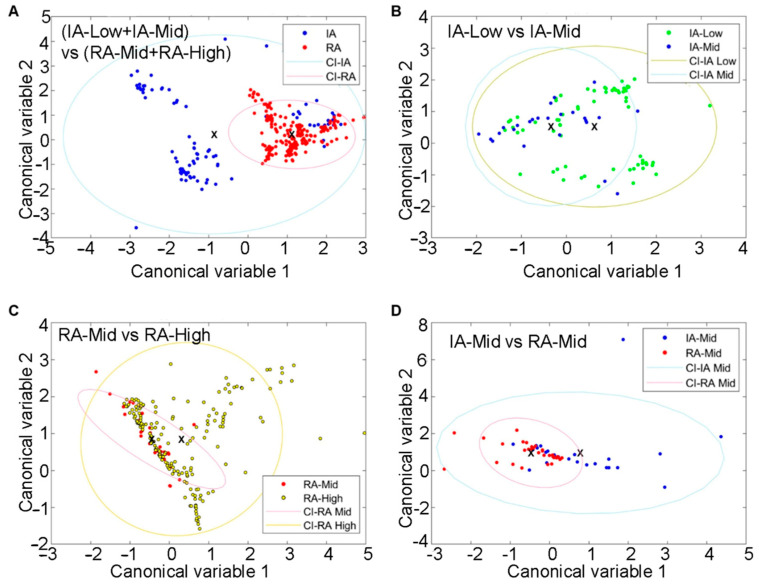
Differentiation of SC from IA and active-RA patients with varying disease and pain states via unmixing and autofluorescence feature analysis. (**A**) Clusters of IA and active cells, (**B**) clusters of IA-Low and IA-mid individuals based on SC, (**C**) clusters of RA-Mid and RA-High cells in the two groups, and (**D**) clustering of IA-Mid and RA-Mid cells. Symbols represent individual cells. Ellipses are one standard deviation around the mean, with the central point indicated by a cross.

**Figure 5 ijms-25-11996-f005:**
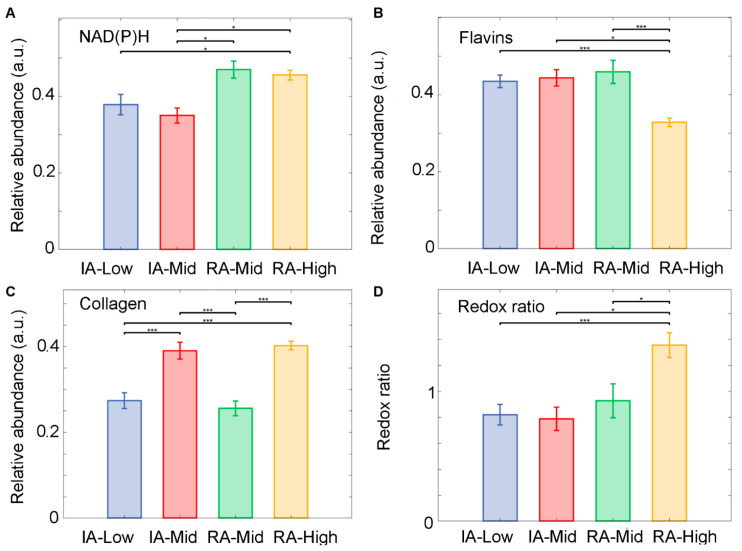
Unmixed autofluorophore abundances for synovial fibrous tissue (FT). Samples were obtained from inactive (IA) and active-RA patients. (**A**) Relative abundance of NAD(P)H, (**B**) relative abundance of flavins, (**C**) relative abundance of collagen, and (**D**) the ratio of relative abundances of NAD(P)H and flavins. Median of the relative abundances displayed and SEM. * *p* < 0.05, and *** *p* < 0.001.

**Figure 6 ijms-25-11996-f006:**
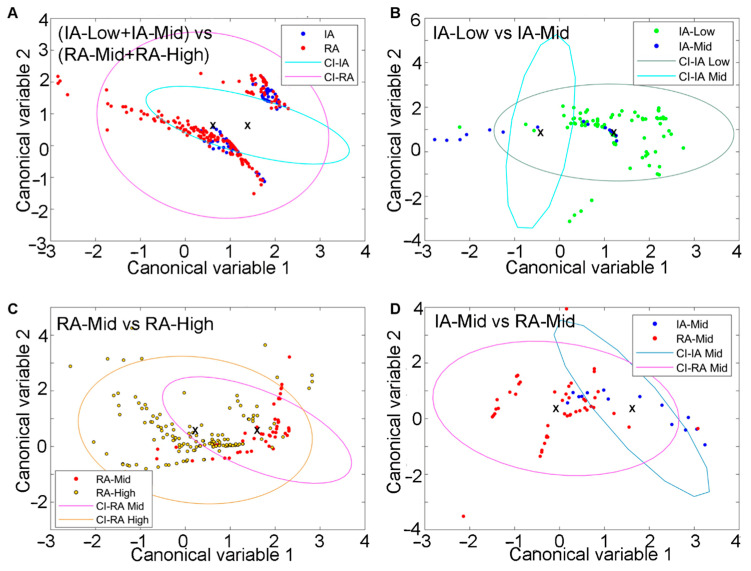
Differentiation of FT from inactive (IA) and active RA patients with varying disease and pain states by using both unmixing and auto fluorescent features. (**A**) Clusters of fibres from IA and active fibres, (**B**) clusters of IA (low and mid) fibres, (**C**) clusters of active (mid and high) fibres in the two groups, and (**D**) clustering of IA-Mid and RA-Mid fibres. Symbols represent individual fibres. Ellipses are one standard deviation around the mean, with the central point indicated by a cross.

**Figure 7 ijms-25-11996-f007:**
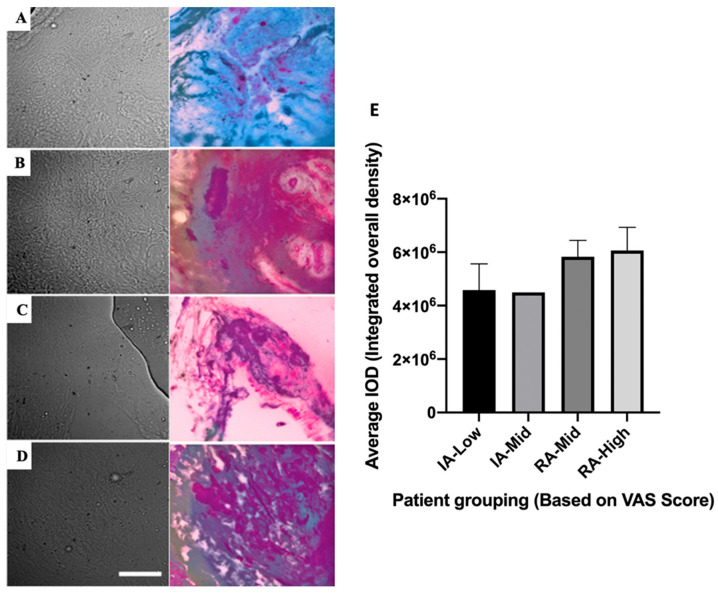
Bright-field images (first column) of synovial tissues with corresponding images highlighting disease state differences (by CYMK colour, second column). Rows are presented for four different groups: (**A**) Inactive (IA)-Low, (**B**) IA-Mid, (**C**) RA-Mid, and (**D**) RA-High. Scale bar is representative of 50 µm. Histological assessment of collagen type II abundance within ST was performed to validate the HAI process (**E**). Error bars represent SEM (IA-Low *n* = 8, IA-Mid *n* = 1, RA-Mid *n* = 4, RA-High *n* = 9).

## Data Availability

The original contributions presented in the study are included in the article/Appendix A; further inquiries can be directed to the corresponding author/s.

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
