# Peer review of "Utilising Hyperspectral Autofluorescence Imaging in the Objective Assessment of Disease State and Pain in Patients with Rheumatoid Arthritis"

_ijms, 2024, doi:10.3390/ijms252211996_

Round 1
Reviewer 1 Report
Comments and Suggestions for Authors
This article presents the results of a study that evaluates the role of utilizing hyperspectral autofluorescence imaging in the objective assessment of disease state and pain in patients with rheumatoid arthritis. The idea is interesting especially since the subject of unmet needs in pain control and evaluation of RA patients is a very hot one.
-row 44 – DAS does not evaluates disease progression! It is a composite score used to measure disease activity!
- please introduce one image in order to explain the role/way of action of Hyperspectral autofluorescence imaging (HAI)
- methodology of the study is not very clear exposed
- patients sample is (almost) not described (number of pts mentioned only on abstract), thereafter just 2 rows (99, 100)
- what about RA comorbidities that could be associated with joint pain (like osteoarthritis and fibromyalgia) – did you screen them/excluded?
Finally I’m very intrigued about 72% match on iThenticate report
Author Response
Reviewer 1
This article presents the results of a study that evaluates the role of utilizing hyperspectral autofluorescence imaging in the objective assessment of disease state and pain in patients with rheumatoid arthritis. The idea is interesting especially since the subject of unmet needs in pain control and evaluation of RA patients is a very hot one.
-row 44 – DAS does not evaluates disease progression! It is a composite score used to measure disease activity!
Thank you for your comment. We agree. This inadvertent error has been corrected. On line 44 “progression” has been replaced with “activity” and the document has been checked to ensure that the error does not reoccur.
- please introduce one image in order to explain the role/way of action of Hyperspectral autofluorescence imaging (HAI)
A new figure explaining hyperspectral imaging has been added to the text as requested.
- methodology of the study is not very clear exposed
The methodology has been revised to improve clarity and detail.
- patients sample is (almost) not described (number of pts mentioned only on abstract), thereafter just 2 rows (99, 100)
The ‘Patient Demographics’ section of the Results has been expanded with additional cohort details. It now reads as follows:
“The study included ST samples from 22 participants with RA (13 female, 9 male). In the inactive group mean DAS28 was 1.8, while for the active group it was 6.3. The mean VAS pain score was 9.0 and 65.5 for patients with inactive and active RA, respectively while AS fatigue was 14.7 and 59.9. The average age of the cohort was 55.4 ± 9.4 years. In the inactive group four out of the nine participants were female, while in the active group it was nine out of thirteen”
- what about RA comorbidities that could be associated with joint pain (like osteoarthritis and fibromyalgia) – did you screen them/excluded?
Inclusion was based on patients fulfilling the 2010 American College of Rheumatology (ACR)/ European Alliance of Associations for Rheumatology (EULAR) criteria. We have now make this more clear in the manuscript, including both the methodology and results.
Finally I’m very intrigued about 72% match on iThenticate report
This manuscript is based on the work carried out by the lead author in fulfilment of her PhD. The iThenticate report is the result of the availability of her thesis.
Reviewer 2 Report
Comments and Suggestions for Authors
The authors present a manuscript analyzing the hyperspectral autofluorescence method to make an objective assessment of pain in patients with rheumatoid arthritis.
Comments:
1) The abstract should be divided into sections (e.g., background, purpose of the study, methods, results and conclusions) for better understanding.
2) Line 15 should specify that swelling and pain refer to joints.
3) DAS28 is not used to assess the progression of RA (line 44) but its activity. By progression is meant the radiological progression. This should be corrected
4) The sample size of patients is small. The authors should comment that this is a limitation of the study.
5) Patients were divided into active and inactive. However, the activity status (mild, moderate, high) was not assessed in this study by calculating the DAS28 value. However, this aspect is important for correct interpretation of the data.
6) The study requires an invasive method to obtain synovial tissue. The authors should comment that this method cannot replace clinical examination with the traditional calculation of DAS28, which allows rapid assessment of activity status and effectiveness of therapy in a busy outpatient clinic.
7) Pain has a subjective component. The authors should better clarify why a histologic assessment is advantageous in clinical evaluation.
8) Although the sample is small, it would be interesting to know whether there are significant differences between male and female patients. It is known that females have a greater perception of pain.
9) The possible presence of fibromyalgia as a co-morbidity should be commented on as a source of pain that can often confuse the physician about the activity status of rheumatoid arthritis.
English should be made a little more fluid for better understanding of the text.
Author Response
Reviewer 2
1) The abstract should be divided into sections (e.g., background, purpose of the study, methods, results and conclusions) for better understanding.
We haven’t divided the abstract into sections since that is the not the requested style of the publishing journal, but would be happy to do so if you feel that will make the message clearer, or keep it as it currently is.
2) Line 15 should specify that swelling and pain refer to joints.
This change has been made as requested.
3) DAS28 is not used to assess the progression of RA (line 44) but its activity. By progression is meant the radiological progression. This should be corrected
Our apologies for the error. The correction has been made as requested and “progression” has been replaced with “activity”.
4) The sample size of patients is small. The authors should comment that this is a limitation of the study.
This issue is addressed in the Discussion as follows:
“Limitations specific to the use of unsupervised machine learning analysis include having a small heterogeneous sample set. Hence, this should be considered a training data set that requires further testing to examine how the model will generalise to additional data sets. A longitudinal clinical cohort study with a larger sample size is required to further validate the use of HAI in RA management. “
5) Patients were divided into active and inactive. However, the activity status (mild, moderate, high) was not assessed in this study by calculating the DAS28 value. However, this aspect is important for correct interpretation of the data.
As noted above, the sample size available to the study was relatively low and would not support greater granularity of assessment. We now note this limitation in the discussion as follows:
“A related limitation is that we were unable to divide activity statuses with greater granularity (e.g. mild moderate and high) which would enable increased precision of findings, nor were sub group analyses (e.g. females in comparison to males) possible. A longitudinal clinical cohort study with a larger sample size is required to further validate the use of HAI in RA management.”
6) The study requires an invasive method to obtain synovial tissue. The authors should comment that this method cannot replace clinical examination with the traditional calculation of DAS28, which allows rapid assessment of activity status and effectiveness of therapy in a busy outpatient clinic.
We have addressed this comment as follows:
“Our findings provide promising support for the feasibility of HAI technology for management and pain assessment, although conventional examination through validated methodologies remains vital.”
7) Pain has a subjective component. The authors should better clarify why a histologic assessment is advantageous in clinical evaluation.
This importance of non-subjective measures is described as follows:
“Traditional studies have used subjective measures, such as PGA VAS, to assess whether a treatment has worked. However, this has the disadvantage that patients may clinically respond (for e.g. with reduction of their swollen joint counts and inflammatory markers) but still not detect a difference (because of ongoing pain impacting their perception of disease activity and the visual analogue score for the patient global assessment), therefore lowering the perceived effectiveness of the treatment on pain.”
8) Although the sample is small, it would be interesting to know whether there are significant differences between male and female patients. It is known that females have a greater perception of pain.
Thank you for your insightful comment. We agree that it is known females have a greater perception of pain. Although it would be desirable to be able to investigate the data set with this level of detail, as you have correctly noted, our sample size is relatively small, and, especially with the unbalanced numbers of males and females in the different groups, statistical assessment had a high propensity to give misleading results. We have therefore updated the discussion to note this issue “nor were sub group analyses (e.g. females in comparison to males) possible.”
9) The possible presence of fibromyalgia as a co-morbidity should be commented on as a source of pain that can often confuse the physician about the activity status of rheumatoid arthritis.
This is now addressed as follows:
“However, this has the disadvantage that patients may clinically respond but still not detect a difference, therefore lowering the perceived effectiveness of the treatment on pain. There is also the potential for confounding from co-morbidities such as fibromyalgia.”
Round 2
Reviewer 2 Report
Comments and Suggestions for Authors
The authors responded appropriately to my comments and modified the text accordingly